# Effects of Circulating HMGB-1 and Histones on Cardiomyocytes–Hemadsorption of These DAMPs as Therapeutic Strategy after Multiple Trauma

**DOI:** 10.3390/jcm9051421

**Published:** 2020-05-11

**Authors:** Birte Weber, Ina Lackner, Meike Baur, Giorgio Fois, Florian Gebhard, Ingo Marzi, Hubert Schrezenmeier, Borna Relja, Miriam Kalbitz

**Affiliations:** 1Department of Traumatology, Hand-, Plastic- and Reconstructive Surgery, Center of Surgery, University of Ulm, 89081 Ulm, Germany; birte.weber@uni-ulm.de (B.W.); ina.lackner@uni-ulm.de (I.L.); meike.baur@uni-ulm.de (M.B.); florian.gebhard@uniklinik-ulm.de (F.G.); 2Institute of General Physiology, University of Ulm, 89081 Ulm, Germany; giorgio.fois@uni-ulm.de; 3Department of Trauma, Hand and Reconstructive Surgery, Goethe University of Frankfurt, 60590 Frankfurt, Germany; marzi@trauma.uni-frankfurt.de (I.M.); info@bornarelja.com (B.R.); 4Institute of Transfusion Medicine, University of Ulm and Institute of Clinical Transfusion Medicine and Immunogenetics Ulm, German Red Cross Blood Transfusion Service Baden-Württemberg—Hessen and University Hospital Ulm, 89081 Ulm, Germany; h.schrezenmeier@blutspende.de; 5Department of Radiology and Nuclear Medicine, Experimental Radiology, Otto-von-Guericke University, 39120 Magdeburg, Germany

**Keywords:** hemadsorption, early myocardial damage (EMD), DAMPs, post-traumatic, calcium handling, mitochondrial dysfunction

## Abstract

Background and purpose: The aim of the study was to determine the effects of post-traumatically released High Mobility Group Box-1 protein (HMGB1) and extracellular histones on cardiomyocytes (CM). We also evaluated a therapeutic option to capture circulating histones after trauma, using a hemadsorption filter to treat CM dysfunction. Experimental Approach: We evaluated cell viability, calcium handling and mitochondrial respiration of human cardiomyocytes in the presence of HMGB-1 and extracellular histones. In a translational approach, a hemadsorption filter was applied to either directly eliminate extracellular histones or to remove them from blood samples obtained from multiple injured patients. Key results: Incubation of human CM with HMGB-1 or histones is associated with changes in calcium handling, a reduction of cell viability and a substantial reduction of the mitochondrial respiratory capacity. Filtrating plasma from injured patients with a hemadsorption filter reduces histone concentration ex vivo and in vitro, depending on dosage. Conclusion and implications: Danger associated molecular patterns such as HMGB-1 and extracellular histones impair human CM in vitro. A hemadsorption filter could be a therapeutic option to reduce high concentrations of histones.

## 1. Introduction

Early myocardial damage (EMD) after trauma is complex and multifactorial. Patients with elevated troponin levels in the emergency rooms have a higher injury severity score (ISS) and require often catecholamines [1,2].

Post-traumatic cardiac injury can be caused by mechanical forces on the heart, particularly in the case of blunt chest trauma [3,4,5] or by inflammation, which mediates secondary cardiac damage. Severe trauma often leads to a systemic inflammatory response, accompanied by the release of danger associated molecular patterns (DAMPs), such as the high mobility group box-1 protein (HMGB-1) and extracellular histones. In humans, the release of HMGB-1 after trauma has been associated with injury severity, the activation of the complement system and increased mortality rates [6].

Extracellular histones have been linked to trauma-induced lung injury [7,8] and to cardiomyopathy during sepsis in mice [9]. In traumatized patients, enhanced levels of circulating histones were correlated with the Sequential Organ Failure Assessment score, endothelial damage and the activation of the coagulation system [8]. In experimental blunt chest trauma in rats and in experimental multiple trauma in pigs and mice, we observed a systemic release of extracellular histones [3,4,5].

HMGB-1 and extracellular histones act via Toll-like receptors (TLRs), which trigger the proinflammatory cytokine signalling, which is known to be cardio-depressive: Histones have been shown to act via TLR 2 and 4 on cardiomyocytes [9]. Furthermore, the role of the TLR9 interaction with histones has been described in liver injury [10]. TLR2 and 4 are activated on neutrophils by HMGB-1 [11]. In mice, the interaction between TLR 4 and HMGB-1 has been described in the development of liver injury [12]. Extracellular histones or HMGB-1 interaction with TLR4 increased the release of inflammatory cytokines such as tumor necrosis factor (TNF), interleukin (IL-)1ß and IL-6, which were shown to be cardio-depressive [13,14,15,16]. Additionally, histones were linked to the release of HMGB-1 via cell damage in liver, lung and kidney injuries in mice [13]. HMGB-1 is known to provoke cardiomyocyte dysfunction in cardiac hypertrophy, heart failure [17] and myocardial ischemia [18]. Extracellular histones are associated with increased intracellular reactive oxygen species, intracellular calcium in rodent cardiomyocytes and reduced mitochondrial membrane potential and ATP concentration [9].

We propose that HMGB-1 and extracellular histones can set off the development of post-traumatic cardiac dysfunction We also investigated whether hemadsorption may provide a therapeutic option to reduce the negative effects of extracellular histones. Previously, it was shown that hemadsorption improved the outcomes of patients with endotoxemia [19,20], necrotizing fasciitis and septic shock [21]. While the resorption of HMGB-1 by hemadsorption filter systems has been described before, we assessed the resorption capacity in regard to extracellular histones.

## 2. Material and Methods

### 2.1. In Vitro Incubation of Human Cardiomyocytes (CMs) with HMGB-1 and Histones

Human iPSC-derived cardiomyocytes (iPSC-CMs) were purchased from Cellular Dynamics, USA. The iPSC-CMs #11713 from Cellular Dynamics were donated from a healthy female donor. The age group at collection was 35–39 years and iPSC-CMs were obtained by episomal reprogramming of PBMC. The cells were cultured for 10 days at 37 °C and 7% CO_2_ in iCell maintenance medium (Cellular Dynamics, Madison, WI, USA). After 10 days cells were incubated with either 100 ng/mL recombinant human HMGB-1 (R&D Systems) or with a mixture of 20 µg/mL extracellular histones (Sigma, St Louis, Missouri, USA) for 6 h at 37 °C and 7% CO_2_. To assess caspase-3/7 activity in treated cardiomyocytes the Caspase-Glo^®^ 3/7 Assay (Promega, Madison, WI, USA) was used. Furthermore, cell viability was measured in the presence of histones or recombinant human HMGB-1 by using the Cell Titer-Glo^®^ Luminescent Cell Viability Assay (Promega, Madison, WI, USA). Additionally, the appearance of HMGB-1 in the supernatant of cardiomyocytes cultured in the presence of histones were detected by using a HMGB-1-ELISA (lBL international, Hamburg, Germany). For all experiments *n* = 6.

### 2.2. In Vitro Incubation of HL-1 Cells with HMGB-1

For in vitro experiments, the murine cardiac muscle cell line (HL-1 cells) (Sigma Aldrich, St. Louis, MO, USA) was used. Murine HL-1 cells were cultured in HL-1 expansion medium at 37 °C in an atmosphere of 5% CO_2_. Following this, HL-1 cells were incubated with HMGB-1 (R&D Systems) for 6 h. Cell viability was detected by using a Cell Titer-Glo® Luminescent Cell Viability Assay (Promega, Madison, WI, USA) in the presence of different concentrations of HMGB-1 (1 µg/mL, 100 ng/mL, 10 ng/mL). Moreover, we analysed the metabolic activity of the cells by MTT assay (Invitrogen, Waltham, MA, USA) in the presence of HMGB-1 dose-dependently (after 24 h of incubation). For all experiments *n* = 6.

### 2.3. Calcium Measurements

For calcium measurements, human CM were incubated with 20 µg/mL histones or with 100 ng/mL HMGB-1 for 60 min before the start of the experiments as well as for the duration of the experiment. To measure changes in intracellular Ca^2+^ concentration, cells were loaded with 5 µM Fura-2 (ThermoScientific, Waltham, MA, USA) for 30 min (in the presence of pharmacological compounds if needed). Fluorescence imaging was performed on a cell observer inverse microscope (Zeiss, Jena, Germany). Cells were illuminated for 90 ms at a rate of 2 Hz at each excitation wavelength (340 and 380 nm). Images were acquired using MetaFluor (Molecular Devices, Ismaning, Germany). Fura-2 ratios were calculated with ImageJ and the data obtained were analysed with the Matlab script PeakCaller [22]. The images were loaded in ImageJ and after background subtraction Fura-2340/380 ratios were calculated. The Fura-2 ratio traces representing changes in cytoplasmic Ca^2+^ concentration were analysed with the Matlab script PeakCaller [22] that allows to obtain the values of rise and decay time and height of calcium transients. For all experiments *n* = 6.

### 2.4. Mitochondrial Respiration

Mitochondrial respiration was analysed by using the Seahorse XFe96 Analyzer (Agilent Technologies, Santa Clara, CA, USA). For this experiment, human cardiomyocytes (iPS) were seeded in special Seahorse XFe96 cell culture plates (Agilent Technologies, Santa Clara, CA, USA) and were cultured for 10 days in iCell maintenance medium (Cellular Dynamics, Madison, WI, USA) at 37 °C and 7% CO2. After the cultivation, cells were treated either with 20 µg/mL extracellular histones or with 100 ng/mL HMGB1 in iCell maintenance medium (Cellular Dynamics, Madison, WI, USA) for 5 h at 37 °C and 7% CO2 and for an additional hour with either 20 µg/mL extracellular histones or with 100 ng/mL HMGB1 in Agilent Seahorse XF DMEM medium pH 7.4 (Agilent Technologies, Santa Clara, CA, USA), supplemented with 1 mM sodium pyruvate (Sigma Aldrich, St. Louis, MO, USA), 2 mM L-Glutamine (ThermoFisher, Waltham, MA, USA) and 50 mM glucose (Sigma Aldrich, St. Louis, MO, USA) at 37 °C and 7% CO2. After exposure of the cells to extracellular histones or HMGB1, the mitochondrial respiration was measured. Therefore, the Seahorse XF Cell Mito Stress Test Kit (Agilent Technologies, Santa Clara, CA, USA) was used. With the Seahorse XF Cell Mito Stress Test mitochondrial function of cells can be assessed and multiple parameters are obtained in one assay, including basal respiration, maximal respiration and spare respiratory capacity. During the entire experimental procedure, the oxygen consumption rate (OCR) in pmol/min is measured continuously by the Seahorse XFe96 Analyzer (Agilent Technologies, Santa Clara, CA, USA). The mitochondrial respiration was assessed as follows: during the experimental process, 2 µM oligomycin, 1 µM carbonyl cyanide 4-(trifluoromethoxy) phenylhydrazone (FCCP) and 0.5 µM antimycin A and rotenone were added to the cells by programmed injection in defined intervals and after each addition the OCR of the cells were measured. After the experiments, the different parameters for basal respiration, maximal respiration and spare respiratory capacity were calculated by using the Seahorse Wave 2.4 software (Agilent Technologies, Santa Clara, CA, USA). Following this, the obtained parameters of mitochondrial respiration were further normalized on the total amount of mitochondria of the cells. Therefore, the cells were fixed with 4% formalin at 4 °C overnight after the experiment. Then, the cells were stained with 0.2% Janus-Green solution, which specifically stains the mitochondria of cells. Afterwards, cells were washed and resolved with 0.5 M hydrochloric acid. Optical density (OD) was measured at 630 nm, correlating with the amount of cellular mitochondria. Then, the oxygen consumption rate (OCR) values of the mitochondrial parameters were normalized to the OD 630 nm values, respectively. Results were evaluated using Seahorse Wave 2.4 software (Agilent Technologies, Santa Clara, CA, USA). For all experiments *n* = 6.

### 2.5. Plasma Samples of Polytrauma Patients

Human plasma samples from 20 multiple injured patients with a history of acute blunt or penetrating trauma and an ISS ≥16 were collected after hospital admission to the University Hospital of the Goethe-University Frankfurt. The plasma collection was approved by the institutional ethics committee (312/10), in accordance with the Declaration of Helsinki. All enrolled patients signed the written informed consent form themselves or written informed consent was obtained from the nominated legally authorized representative on the behalf of participants in accordance with ethical standards. Exclusion criteria were younger than 18 or older than 80 years of age, severe burn injury, acute myocardial stroke, cancer or chemotherapy, immunosuppressive drug therapy, HIV, infectious hepatitis, acute cytomegalovirus (CMV) infection and/or thromboembolic events.

Blood samples were withdrawn in ethylenediaminetetraacetic acid (EDTA) tubes (Sarstedt, Nürmbrecht, Germany) directly after admission. The samples were kept on ice until centrifugation at 2100× *g* for 15 min. Then the supernatant was collected and stored at −80 °C until assay.

### 2.6. Hemadsorption in Plasma of Multiple Injured Patients

We developed small hemadsorption-columns for a volume of 150 µL plasma by using a Cytosorb^®^ 300-column (CytoSorbents Inc., Monmouth Junction, NJ, USA). 300 µL of the content of the column was aliquoted in Ultrafiltration Spin-Columns (0.45 cutoff; Merck Milipore, Billerica, MA, USA) and was centrifuged before incubation with plasma of multiple trauma patients. After 6 h of incubation an additional centrifugation followed. The samples were then stored at −80 °C. Furthermore, we investigated a dose-depending hemadsorption-curve of histones (mixture of histones). Therefore, histones of calf thymus were diluted in Aqua dest. with the concentrations 700, 500, 300, 200, 100, 50 and 25 µg/mL. They were incubated for 6 h on a plate shaker and were centrifuged before measuring the histone concentrations.

### 2.7. Statistical Procedures

All values are expressed as mean ± SEM. Data were analysed by one-way ANOVA followed by Dunnett’s or Tukey’s multiple comparison test. Students *t*-test was used in the case of comparison of two groups. *p* ≤ 0.05 is considered statistically significant. GraphPad Prism 7.0 software was used for statistical analysis (GraphPad Software, Incorporated, San Diego, CA, USA).

## 3. Results

### 3.1. Decreased Apoptosis in Human Cardiomyocytes and Alterations of Calcium Handling in Presence of HMGB-1 and Histones

In the presence of histones, cell viability of cardiomyocytes was significantly reduced (Figure 1A), whereas the detection of pro-apoptotic caspase was neither changed in the histone nor in the HMGB-1 treated CM (Figure 1B). We also investigated changes in calcium handling of CM in the presence of histones or HMGB1. The mean height calcium signal was reduced in the presence of HMGB-1 (Figure 1C), while the mean rise time of the calcium signal was neither changed in the presence of histones nor after incubation with HMGB-1 (Figure 1D). In Figure 1E, we demonstrate the increase of the mean decay time of the calcium signal in presence of histones. Furthermore, the frequency of the calcium signal in CM was significant reduced in the presence of histones, as well as after incubation with HMGB-1, which was associated with bradycardia of the CMs in vitro (Figure 1F).

### 3.2. Decrease in HL-1 Cell Viability and Metabolic Activity in Presence of Different HMGB-1 Concentrations

Cell viability of HL-1 cells were significantly reduced in the presence of 1 µg/mL, 100 ng/mL as well as 10 ng/mL HMGB-1 (Figure 2A). By conducting the MTT assay, we detected a reduction of metabolic activity in the HL-cells in the presence of all tested HMGB-1 concentrations (Figure 2B).

### 3.3. Metabolic Alterations of Human Cardiomyocytes in Presence of Histones

Metabolic alterations of human CMs in the presence of a mixture of histones or HMGB-1 were investigated. The basal respiration decreased tendentially in the presence of histones or HMGB1 compared to the control group (Figure 3A,D). Moreover, we observed a decrease of the maximal respiration capacity in the presence of both HMGB-1 and histones (Figure 3B,E). In addition, the spare respiratory capacity decreased significantly in the presence of histones or HMGB-1 (Figure 3C,F).

### 3.4. Hemadsorption—A Therapeutic Option to Eliminate Systemic Extracellular Histones

To evaluate the therapeutic potential of hemadsorption we analysed the absorption capability and capacity for extracellular histones by CytoSorb® 300. After incubation of different histone concentrations, the levels decreased between 92% and 99% within 6 h (Figure 4A). Moreover, extracellular histone concentration in blood samples of multiple injured patients collected at admission to the hospital significantly dropped after absorption by the hemadsorption filter (Figure 4B).

## 4. Discussion

In this report, we demonstrated the detrimental effects of HMGB-1 and histones on cardiomyocytes. Recent studies revealed an increase of circulatory histones after experimental blunt chest trauma in rats [4], after multiple trauma in pigs [3] and of circulating HMGB-1 in pigs after multiple trauma [23]; observed together with depressed cardiac function [3]. The presented data underpins that both HMGB-1 and extracellular histones impair myocardial functions, leading to consequences such as a deceleration of frequency of CMs and an impaired mitochondrial respiration in vitro. 

The role of HMGB1 on cardiac function and the potential detrimental effects are controversial so far. On the one hand, HMGB-1 is systemically elevated in models of myocarditis and ischemic myocardial infarction in mice [24]. In addition, the application of anti-HMGB-1 antibody in hemorrhagic shock in mice was associated with decreased systemic release of cardiac enzymes, reduced local ATP depletion and systemic levels of inflammatory mediators like TNF and IL-1ß [25]. Extracellular HMGB-1 binds directly to TLR 4 as well as to the receptor for advanced glycation end products (RAGE), which was linked to the production and release of inflammatory cytokines [14,15] such as TNF, IL-1ß and IL-6 [16], which were shown to be cardio-depressive [26]. An increase of systemic IL-6 concentration has been linked to cardiac dysfunction, demonstrated as a reduction of stroke volume, cardiac output and the performance of the left ventricle [27,28]. In rat CMs the presence of TNF and IL-1ß led to dysfunction of the calcium balance: it prolongs the calcium transient duration and therefore the action potential, as well as leads to asynchronous calcium release during electrical stimulation. Further, these cytokines increased the vulnerability of the sarcoplasmic reticulum for spontaneous calcium leakage. TNF and IL-1ß depressed calcium transient, the contractility and therefore have been linked to arrhythmogenicity in ventricular rat CM [29]. The presence of IL-1ß could be associated with prolonged action potential duration, the reduction of the transient potassium current of 35%, therefore a reduced repolarization in CM and again the increase of diastolic sarcoplasmic calcium leakage. Together, these changes led to high potential of cardiac arrythmias [30]. Furthermore, RAGE-knock out mice showed lower myocardial inflammation and fibrosis compared to wild-type mice in a model of inflammatory heart disease [31]. Besides, these effects HMGB-1 was shown to influence the post-traumatic development of microvascular thrombosis and endothelial cell activation via inhibition of the anticoagulant protein C pathway mediated by the thrombin-thrombomodulin complex, and further stimulated tissue factor expression on monocytes [32]. High plasma levels of HMGB1 are associated with increased complement activity as indicated by elevated soluble C5b-9 plasma levels that are generated during the late phase of complement activation [6]. The connection between complement activation and cardiac depression is well described after trauma [33] and sepsis [34,35].

On the other hand, Limana, et al. (2005) [36] injected low-dose HMGB-1 in the left ventricles of infarcted mice hearts and observed a partial reconstitution of the defect by a myocardial population after one week compared to the beginning of scar formation in non-HMGB-1 treated animals. The intervention with HMGB-1 resulted in the formation of new myocytes within the infarcted portion of the wall via the proliferation and differentiation of endogenous cardiac c-kit+ progenitor cells [36]. In addition, ejection fraction measured by echocardiography improved in the locally HMGB-1 treated animals after myocardial infarction [36]. The application of HMGB-1 after myocardial infarction also decreased levels of IL-1, IL-6, IL-10 and vascular endothelial growth factor (VEGF) [37]. Transgenic mice with an overexpression of cardiac-specific HMGB-1 were protected against the consequences of myocardial infarction [38], whereas animals with HMGB1 null mutation are nonviable [39].

However, our in vitro data demonstrates a deceleration of the natural frequency of human CMs, as well as changes in the calcium handling in the presence of HMGB-1 and a reduction in mitochondrial respiration. This clearly indicates the negative effects of HMGB-1.

Therefore, therapeutic elimination of high systemic concentrations of HMGB-1 might be an option to reduce cardiac dysfunction after trauma. microRNA-26a inhibited the HMGB-1 expression in an ischemic/reperfusion model, which is correlated with less infiltration of inflammatory cells and a decreased cytokine release [40]. Moreover, an overexpression of microRNA-142-3p targeting HMGB-1 gene in mouse cardiomyocytes presented a significantly lower apoptotic rate as controls after [41]. However, the context of trauma cardiac damage is complex and multifactorial. Therefore, a multipotent therapeutic option could be a hemadsorption filter, which was described to be useful for the elimination of not only HMGB-1 [42] but also other inflammatory molecules such as cytokines, midkine or active complement components [43,44]. The presented report demonstrates the initial usefulness of hemadsorption filters to eliminate extracellular histones. Further studies need to evaluate the therapeutic clinical use after trauma.

Recently circulating histones were discussed as new biomarkers after trauma [45,46]. In previous studies we demonstrated that histones bind to the surface of rat CMs [9]. Incubation of human cardiomyocytes with histones increased the levels of cytosolic reactive oxygen species (ROS) [9]. This observation was dose-dependent [9]. Increased intracellular ROS levels were associated with increased cytosolic calcium concentration in CMs by modulating calcium handling proteins [47] and by blocking the sarcoplasmic endoplasmic reticulum-transporting ATPase (SERCA) [48]. The data revealed that calcium signalling in human CMs was disturbed in the presence of histones and HMGB-1. The frequency of calcium signals decreased either in the presence of HMGB-1 or histones. This was in line with findings obtained from Langendorff perfused mouse hearts in the presence of histones, which showed sinus bradycardia as well as the development of ventricular bigeminy in the ECG [9]. Moreover, this report shows that the mean decay time of the single calcium peaks were altered in human CMs in the presence of histones, which was in accordance with earlier findings demonstrating disturbed calcium handling by enhanced build-up of (Ca^2+^) [34]. Apart from that, it is also known that histones interact with the phospholipid-membrane of cells, which leads to higher permeability and a calcium influx in cells [8,49,50,51]. Rat CM incubated with histones featured increased intracellular calcium, which was demonstrated to be dependent on the expression of TLR2 and 4. In the absence of TLR2 or 4 increases of intracellular calcium in cardiomyocytes in the presence of extracellular histones were ameliorated [9].

Increased intracellular calcium concentration have been linked to the cell toxicity of extracellular histones [8]. Alhamdi et al. (2015) [52] cultured cardiomyocytes with the plasma of patients with sepsis, resulting in a significant reduction of cell viability after incubation with serum containing > 75 µg/mL histones compared to incubation with the sera of healthy controls [52]. In the present study, the cell viability of human cardiomyocytes was decreased in the presence of 20 µg/mL extracellular histones. The caspase 3/7 activity did not change compared to the controls.

As presented in Figure 1, the presence of histone led to changes in mitochondrial respiration, especially the spare respiratory capacity, which was impaired. These findings were in accordance with earlier studies demonstrating reduced mitochondrial membrane potential in isolated rat CMs in the presence of extracellular histones [9].

Therapeutically, anti-histone antibodies have been applied experimentally to reduce high systemic histone concentrations [52,53]. Furthermore, the application of anti-histone antibodies improved cardiac dysfunction in mice with cecal ligation and puncture induced sepsis [9,52]. Hemadsorption is routinely used in critically ill patients [19,20,21]. Here, we proved for the first time that this filter system reduced high concentrations of extracellular histones. Therefore, the hemadsorption is a promising therapeutic option in severely injured patients because of its multifactorial character. It has been shown to not only reduce extracellular histones and HMGB1 but also complement factor C5a and cytokines [19,20,21,42]. Hemadsorption may have to mitigate the consequences of these factors that have all been linked to cardiac damage and cardiac dysfunction. In further clinical studies, the advantages of hemadsorption in multiple injured patients has to be evaluated in regard to its ability to ameliorate EMD.

One major limitation of the present study is the missing testing of CMs function in the presence of a patient’s plasma before and after hemadsorption. A potential depression of CMs function could be examined by analysis of the mitochondrial respiration (basal, spare and maximal respiration) as well as of the CMs contraction in the presence of non-filtered and filtered polytrauma plasma. Furthermore, the positive effects of hemadsorption could be investigated by intracellular calcium measurements, including the analysis of the mRNA expression of important calcium-handling proteins (NCX or SERCA). Further experiments could also include structural changes in human CMs in the presence of filtered and non-filtered polytrauma plasma, for example the measurement of connexins, intracellular troponin, alpha-actinin or desmin. Although these experiments seem interesting and would fit well in the context of the manuscript, they are beyond its scope.

The focus of the present report lies in the distinct effects of HMGB-1 and extracellular histones on human CMs functionality. In a previous study, we treated human CMs with a defined polytrauma-cocktail, including miscellaneous inflammatory cytokines, complement activation products and DAMPs (IL-1ß, IL-6, IL-8, TNF, C5a, C3a, HMGB1 and extracellular histones), mimicking the inflammatory conditions of multiple trauma in vitro [54]. In this study, we demonstrated that the entirety of these inflammatory molecules acted detrimentally on the CMs, impairing their cellular glucose and fatty acid transport, which might contribute to impaired cardiac function after multiple trauma [55]. However, our main focus lies on the specific effects of the individual components, therefore we treated the cells either with HMGB1 or with extracellular histones in the present study. Moreover, we did not use the plasma before and after hemadsorption, because we cannot exclude that the effects on CMs were mediated via other cytokines or DAMPs. The hemadsorption filter system was shown to reduce unpacifically the concentration of a wide range of inflammatory- and damaging molecules, therefore it is used for treatment of patients with SIRS or sepsis. In order to understand the complex pathomechanism of post-traumatic cardiac dysfunction, the particular influence of the individual DAMPs on CMs should be tested primarily as well as the filtration capacity of the hemadsorption column for these specific molecules. Moreover, in order to further verify the distinct effects of the single components, neutralizing antibodies might be used. However, this was not the goal of the present study and should be investigated in detail in future studies. A correct assignment between damaging mediators, their effects on CMs and the benefit of a mediator’s reduction is only possible due to precise testing of the individual mediators.

## 5. Conclusions

To summarise, this study confirms that nuclear proteins provoke cardiac damage and cardiac dysfunction after trauma. HMGB-1 and extracellular histones are released after multiple trauma. Despite their positive roles in the immune reaction, these mediators are associated with detrimental effects on cardiomyocytes including their viability, calcium handling and mitochondrial function. The study also shows that hemadsorption is a useful therapeutic option to reduce DAMPs concentrations and to potentially ameliorate EMD after trauma.

## Figures and Tables

**Figure 1 jcm-09-01421-f001:**
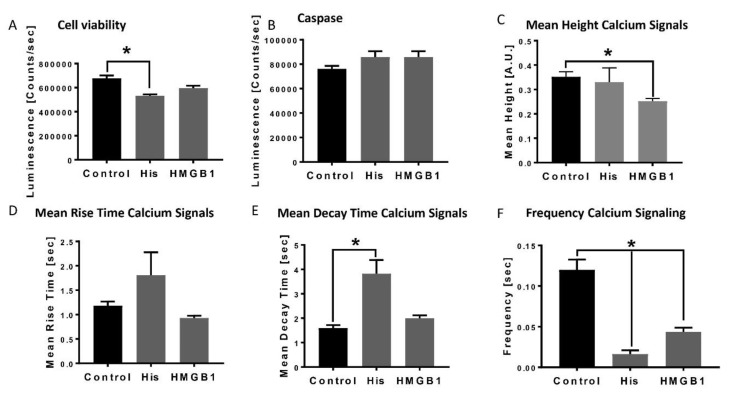
Decrease in cardiomyocytes viability, apoptosis and alterations of calcium handling in the presence of HMGB-1 and histones: Cell viability of human cardiomyocytes (Luminescence in counts/sec) in the presence of 20 µg/mL histones (His) and 100 ng/mL HMGB-1 (37 °C, 6 h) compared to the control (PBS) (**A**). Caspase-3/7 activity (counts/sec) in human CMs in the presence of histones and HMGB-1 (**B**). Mean Height calcium signals (A.U) in the presence of histones and HMGB-1 compared to the control (**C**). Mean rise time (sec) (**D**) and mean decay time (sec) (**E**) of calcium peaks of human CMs in the presence of histones and HMGB-1. Frequency of calcium signals (sec) in human cardiomyocytes treated 20 µg/mL histones and 100 ng/mL HMGB-1 (**F**). Results are presented as mean ± SEM, for all experiments *n* = 6. Indicated results were significant * *p* < 0.05.

**Figure 2 jcm-09-01421-f002:**
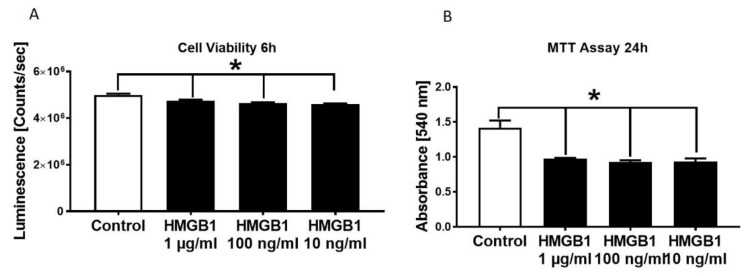
Decrease in HL-1 cell viability and metabolic activity in the presence of different HMGB-1 concentrations. HL-1 cell viability in the presence of different HMGB-1 concentrations after 6 h of incubation compared to control-HL-1 cell viability (**A**). Metabolic activity measured in HL-1 cells in the presence of different HMGB-1 concentrations incubated for 24 h compared to controls (**B**). *n* = 6. Results were significant * *p* < 0.05.

**Figure 3 jcm-09-01421-f003:**
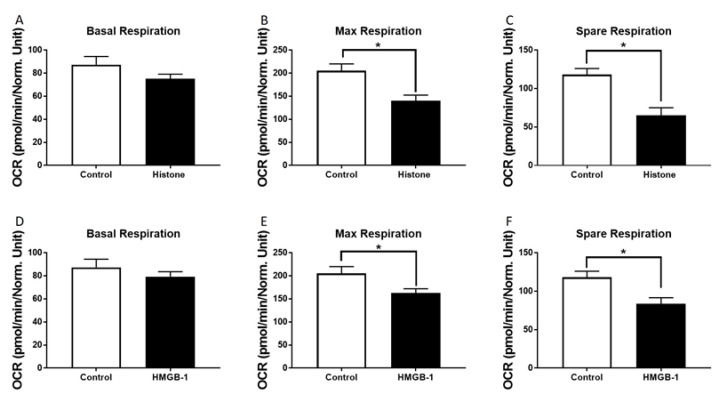
Metabolic alterations of human cardiomyocytes (CM) in the presence of histones: Basal respiration (OCR in pmol/m in/Norm. unit) of human CMs in the presence of 20 µg/mL histones (37 °C, 6 h) (**A**). Maximal respiratory capacity (OCR in pmol/m in/Norm. unit) of human CMs in the presence of 20 µg/mL histones compared to the control (PBS) (**B**). Spare respiration (OCR in pmol/m in/Norm. unit) of human CMs in the presence of 20 µg/mL histones (37 °C, 6 h) (**C**). Basal respiration (**D**), max respiratory capacity (**E**) and spare respiration (**F**) of human CM in the presence of 100 ng/mL HMGB-1. Results are presented as mean SEM, for all experiments *n* = 6. Results are significant * *p* < 0.05, statistical analysis with *t*-test.

**Figure 4 jcm-09-01421-f004:**
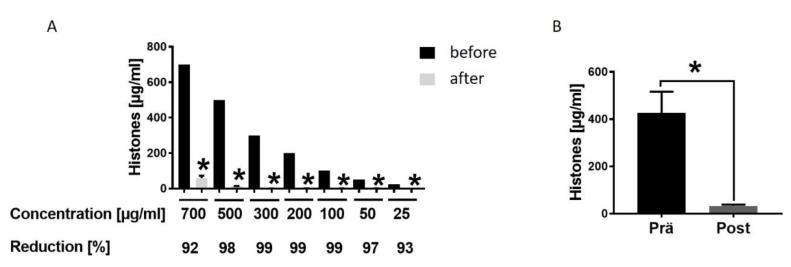
Hemadsorption—a therapeutic option to eliminate systemic extracellular histones. (**A**). Hemadsoption of different concentrations of histones (700, 500, 300, 200, 100, 50, 25 µg/mL) presented as percentage of reduction (%) after 6 hrs incubation time; for all concentrations *n* = 6. (**B**). Histone levels in blood plasma of 22 multiple injured humans (µg/mL) in the Emergency room (pre) and after hemadsorption for 6 h (post). Results were presented as SEM, * *p* < 0.05.

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
