# Peer review of "Effects of Circulating HMGB-1 and Histones on Cardiomyocytes–Hemadsorption of These DAMPs as Therapeutic Strategy after Multiple Trauma"

_jcm, 2020, doi:10.3390/jcm9051421_

Round 1

Reviewer 1 Report

Also I have attached a review file. The following is a\the same as described here.

Authors investigated whether HMGB1 and histone directly influenced cardiomyocyte functions including metabolism and viability and they found that both played a critical role in impairing each feature. As a therapeutic tool to eliminate systemic extracellular histones, they showed that hemadsorption worked well to do so.

For the reviewer, this study is important to provide us a concept that those aggravating factors can be eliminated using such a device in order to prevent cells from exposure of those factors.

However, they need in several parts to revise the original manuscript due to the following concerns and comments;

(major comments)

  1. In the method sections, they need to provide more information regarding calcium measurement. In results authors demonstrated several features of calcium experiments, including mean height calcium, mean rise time, mean decay time, and frequency; however, they did not clearly mention about these parameters, in which they measured. Therefore, they need to add explanations about those in the method section.

 Moreover, in mitochondrial respiration section in the method section, they also need to provide information how OD630 represented OCR. To do so, they used the specific kit, but they should mention about the mechanisms in which OCR can be evaluated by measurement OD630. What did OD630 mean?

  1. The demonstrated that hemadsorption efficiently decreased plasma histones. Did they check whether the blood samples from patients, subjected to hemadsorption, showed less cell (cardiomyocytes) toxicity compared to pre-adsorption blood samples in vitro? They need to show the data.
  2. In the discussion section, they showed a double-edged sword of HMGB-1 effects.

Usually, HMGB-1 has been shown to possess pathological effects. On the other hand, low-dose HMGB-1 shows rather a beneficial effect. They need to discuss more extensively in the discussion section the mechanisms producing differences.

(a minor comment)

They need to check more seriously the manuscript in terms of English grammar.

Reviewer 2 Report

The authors correctly highlight the negative consequences of histone and HMGB-1 on cardiac myocyte viability, calcium handling, and metabolism. Both are elevated well known to have a negative impact on cell function and viability after injury and significant inflammatory response. 

The authors propose an approach to limit histone dependent damage by removing histone from plasma by hemadsorption. They are able to effectively remove histone from plasma in test and patient samples.

Demonstrating the removal of histones is elementary and necessary, however, the authors would greatly improve the impact of this study by using plasma before and after hemadsorption in assays to define effects on cardiac myocytes in Figures 1,2, and 3 to demonstrate potential therapeutic benefit with decreased injury and cell stress.

Reviewer 3 Report

In this manuscript, the authors investigated the effects of HMGB-1 and histones on cardiomyocytes and demonstrated hemadsorption as a potential therapeutic strategy after multiple trauma. They evaluated cell viability, calcium handling, and mitochondrial respiration in human iPSC-derived cardiomyocytes (iPSC-CMs) treated with HMGB-1 and extracellular histones. They also found filtrating plasma from injured patients with a hemadsorption could reduce histone concentration. The experimental design is straightforward and outcomes are clearly illustrated. I have a few comments below.

Major Concerns

  1. The authors should revise the title. It appears these two experimental settings (HMGB-1/histones and hemadsorption) are relatively separate. They should make the title with coherence as a whole theme.
  2. Page 2 Line 68: these should be iPSC-derived cardiomyocytes (iPSC-CMs) instead of “iPS”. As these cardiomyocytes were purchased from Cellular Dynamics, the original patient information should be stated. Are these iPSC-CMs derived from healthy patients? Or patients with certain cardiovascular disease?
  3. The manuscript is relatively descriptive and apparently short of mechanistic insights. Not sure how the concentrations of histones (20 ug/ml) and HMGB-1 (100 ng/ml) were selected. Are these physiological concentrations in human patients? The authors should explain why they chose these concentrations.
  4. Page 6 Line 171-182: it is known hemadsoption can eliminate extracellular histones from blood serum. What is the logistic connection between this experiment and iPSC-CMs experiment? The authors may treat iPSC-CMs with blood serum before and after hemadsoption to test their potential therapeutic effects.
  5. The authors purchased iPSC-CMs from Cellular Dynamics. How many biological replicates did they carry out (n=? patients)? Different patient iPSC-CMs may have distinct responses to HMGB-1 and histone treatment. It would be more interesting if the authors can obtain cardiomyocytes from both healthy controls and patients with cardiac injury to test their respective reactions to HMGB-1 and histones.

Round 2

Reviewer 1 Report

Authors sincerely responded to the referee's concerns and comments.

The referee has nothing to be added further.

Author Response

Thank you for the positive evaluation of our revised manuscript. Please see the attachment.

Reviewer 2 Report

The authors were unable to address the major concern regarding the testing and demonstration of alteration of patient plasma after hemadsorption on myocyte function. I realize the current state during COVID19 exacerbates our research efforts, however, issues relating to testing samples before and after hemadsorption and how to address other factors that are removed is essential for the publication of this manuscript.

Reviewer 3 Report

The authors have addressed my concerns appropriately. 

Author Response

We are thankful for the helpful comments and the positive feedback to our
revised manuscript version. Please see the attachment.
